# Beyond Visual Reconstruction Quality: Object Perception-aware 3D Gaussian Splatting for Autonomous Driving

**Renzhi Wang**
University of Alberta
VCIP, CS, Nankai University

**Yuxiang Fu**
Nankai University

**Wuqi Wang**  **Haigen Min**
Chang'an University

**Wei Feng**
Tianjin University

**Lei Ma**
University of Tokyo
University of Alberta

**Qing Guo** [*]
NKIARI, Shenzhen Futian
VCIP, CS, Nankai University

## Abstract

Reconstruction techniques, such as 3D Gaussian Splatting (3DGS), are increasingly used to generate scenarios for autonomous driving system (ADS) research. Existing 3DGS-based approaches for autonomous-driving scenario generation have, through various optimizations, achieved high visual similarity in reconstructed scenes. However, this route is built on a strong assumption: that higher scene similarity directly translates into better preservation of ADS behaviour. Unfortunately, this assumption has not been effectively validated, and ADS behaviour is more closely related to objects within the field of view rather than the global image. Thus, we focus on the perception module—the entry point of ADS. Preliminary experiments reveal that although current methods can produce reconstructions with high overall similarity, they often fail to ensure that the perception module outputs remain consistent with those obtained from the original images. Such a limitation can significantly harm the applicability of reconstruction in the ADS domain. To address this gap, we propose two complementary solutions: a perception-aligned loss, which directly leverages output differences between reconstructed and ground-truth images during training; and an object zone quality loss, which specifically reinforces training on object locations identified by the perception model on ground-truth images. Experiments demonstrate that both of our methods improve the ability of reconstructed scenes to maintain consistency between the perception module outputs and the ground-truth inputs. We release code at: https://github.com/Shanicky-RenzhiWang/Perception-aware-3DGS

## 1 Introduction

3D Gaussian Splatting (3DGS) is an efficient 3D reconstruction technique that has rapidly advanced in recent years (Kerbl et al., 2023). By enabling photorealistic scene reconstruction from multi-view images, it can accurately capture complex details, directly meeting the scene generation requirements in the field of autonomous driving (Zhou et al., 2024; Peng et al., 2025a; Yang et al., 2023b).

Existing 3DGS methods for traffic scenario generation in autonomous driving still follow optimization objectives similar to those in other domains. Specifically, they focus on improving overall image similarity through metrics such as SSIM, PSNR, and LPIPS (Yan et al., 2024; Zhou et al., 2024; Peng et al., 2025b). However, in the context of autonomous driving, the goal is for ADS to make the same decisions and take the same actions in reconstructed scenes as it would in the original scenes. Only such reconstructions are truly useful for system development and testing. Consequently, the objectives of existing works rely on a strong implicit assumption: that higher image similarity will lead to more consistent behaviour from the autonomous driving system.

---

[*]Corresponding author: tsingqguo@ieee.org

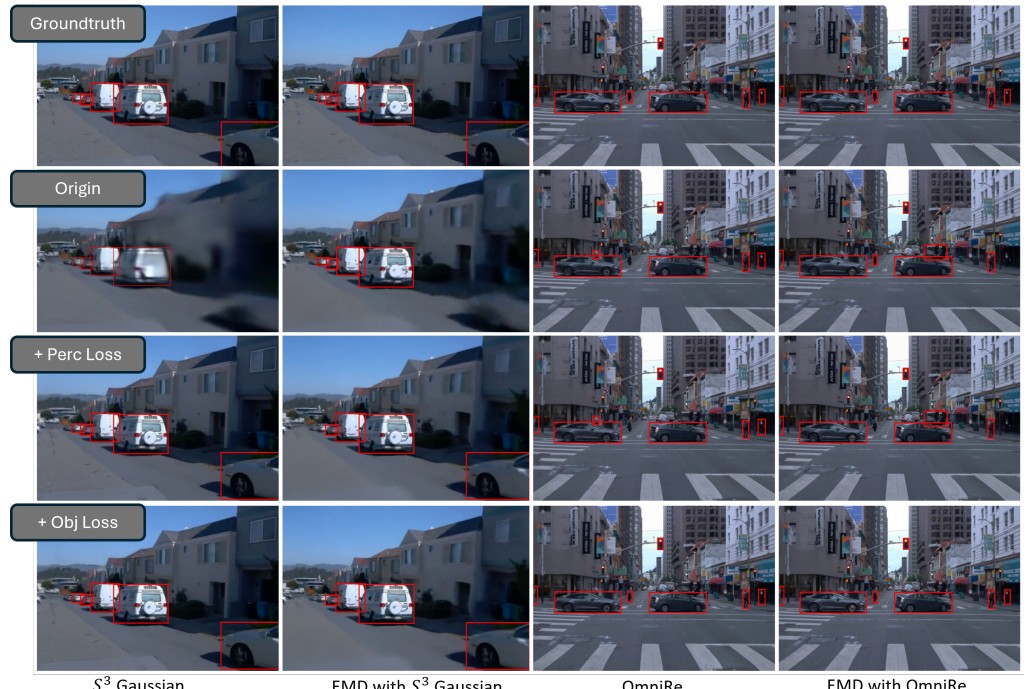

Figure 1: We enhance perception stability in 3D reconstruction by introducing perception-based loss and object zone quality loss, ensuring that the reconstructed images produce perception results consistent with the ground truth for reliable autonomous driving applications. The red box is the bounding box detected by the perception model.

However, this assumption presents a notable gap in the context of autonomous driving. In ADS, the positions, scales, and categories of objects within the sensor's field of view—especially non-player characters (NPCs)—have the most significant influence on system behaviour. Yet these objects often occupy only a small portion of the frame or scene, leading global similarity metrics to under-emphasize their importance during reconstruction. Some prior works (Huang et al., 2024; Wei et al., 2025) have also recognized this issue and attempted to specifically enhance NPC characteristics, thereby partially improving perception recognition of NPCs in reconstructed scenes. Nevertheless, these approaches do not directly address the underlying gap. Even with ground-truth inputs, perception modules can make errors. Therefore, it is not enough to ensure that reconstructed scenes reproduce only the correctly recognized objects; we also need to reproduce the errors that already exist. For example, as shown in the Figure 1, the reconstruction produced by the EMD(+OmnirRe) method can even enable the perception model to detect more objects than in the ground-truth input. However, this is not necessarily a desirable outcome for developers who aim to enhance perception performance through scene reconstruction. Reproducing existing errors is, in fact, aligned with our broader goal for reconstruction techniques in ADS: to efficiently reveal the limitations and issues of ADS perception, enabling more effective system testing and improvement.

This gap inspires us to move beyond the quality of visual reconstruction. We introduce the concept of perception stability, which requires that the same perception model produce consistent outputs across both ground-truth and reconstructed images. By adopting perception stability as an optimization objective, reconstruction methods can better align with the ultimate goal of supporting ADS development and testing.

To validate this gap, we first conduct a preliminary study of multiple 3DGS methods for ADS traffic scene reconstruction, using different perception models. The results show that while existing methods achieve high scores on visual metrics, these improvements do not translate into higher perception stability. This observation highlights the limitations of optimizing solely for visual metrics and motivates us to design improvements with perception stability as the objective. To this end, as shown in Figure 2, we propose two approaches to improve perception stability during reconstruc-

tion. The first directly aligns reconstructions with perception outputs by penalizing inconsistencies in object detection results. The second focus is on the visual fidelity of task-relevant object regions, encouraging accurate reconstruction of object zones detected by the perception model, even if the detection is incorrect.

Through extensive experiments, we demonstrate that both of our proposed improvements not only significantly enhance the perception stability of 3DGS reconstructions—ensuring that perception module outputs on reconstructed scenes remain consistent with those on the original images—but also maintain, or in some cases even improve, the overall visual quality. This highlights that it is possible to design reconstruction methods that are both visually accurate and functionally meaningful, supporting the development and testing of autonomous driving systems.

In summary, this paper makes the following contributions:

- We introduce perception-aware reconstruction, a principle that reconstructed scenes should maintain visual realism while preserving perceptual stability.
- We propose two approaches: a perception-aligned loss and an object-zone quality loss, which could improve perception stability when integrated into the training process.
- We validate our methods using large-scale experiments, demonstrating that they improve perception stability without compromising overall visual quality.

More broadly, the proposed perception-aware reconstruction principle, along with the two methods we introduce, establishes a foundation for task-consistent reconstruction in other domains, such as robotics and AR/VR, which require perception outputs as the basis for downstream modules. Our approach demonstrates how reconstruction can be made both visually realistic and practically helpful for real-world applications.

## 2 BACKGROUND

**Traffic Scene Reconstruction** Traffic scenario generation is an important research direction for autonomous driving development and testing, while reconstruction techniques enable more flexible and realistic scenarios (Nalic et al., 2020; Zhong et al., 2021; Wu et al., 2024). In the early stages of research, a series of NeRF-based works (Mildenhall et al., 2021; Luo et al., 2023), such as Block-NeRF (Tancik et al., 2022), Mega-NeRF (Turki et al., 2022), and EmerNeRF (Yang et al., 2023a), explored and attempted to capture the characteristics of traffic scenes. With the emergence of 3DGS, techniques represented by StreetGaussian (Yan et al., 2024), DrivingGaussian (Zhou et al., 2024), and DeSiRe-GS (Peng et al., 2025a) have rapidly proliferated. To date, methods like $S^3$Gaussian (Huang et al., 2024), OmniRe (Chen et al., 2025), and EMD (Wei et al., 2025) lead the field in terms of reconstruction performance and scalability.

**Perception Robustness in AI Systems** As the first stage in many AI systems, perception plays a critical role, and its robustness and stability are essential for overall system reliability. Dong et al. (2023) proposed a robustness benchmark for autonomous driving system (ADS) perception modules based on Waymo (Sun et al., 2020), KITTI (Geiger et al., 2013), and nuScenes (Caesar et al., 2020) datasets, finding that motion-level noise has the most significant impact on perception robustness. NLTE (Liu et al., 2022) highlighted the importance of interactions between image noise and multi-scale features for perception stability. Gupta et al. (2024) further showed that changes in weather conditions can also pose significant challenges to object detection results. Additionally, works such as Song et al. (2024) et al. have jointly evaluated perception models in terms of accuracy, latency, and robustness, revealing that current methods still leave substantial room for improvement in safe and stable operation. Overall, perception robustness remains an unsolved challenge. Therefore, for reconstruction tasks, minimizing noise that could affect perception outcomes is crucial and a necessary condition for practical deployment.

## 3 PRELIMINARY STUDY

### 3.1 PROBLEM DEFINITION

Given an input scene $S$, a 3DGS model $\mathcal{R}$ reconstruct a 3D representation of the scene. A perception module(in this work, specifically a detection model, since it is generally considered a core and

indispensable perception module in ADS systems (Gog et al., 2021; Wang et al., 2025)) $\mathcal{P}$ produces necessary outputs for downstream modules in ADS. In typical 3DGS tasks, the goal of reconstruction is typically to minimize the difference between the ground truth image $x \in S$ and the rendered image of the reconstructed model $\mathcal{R}(x)$. This objective can be expressed as

$$\mathcal{L}_{\text{recon}} = d_{\text{img}}(\mathcal{R}(x), x) \tag{1}$$

where $d_{img}(\cdot, \cdot)$ measures the visual quality(e.g. SSIM, PSNR).

However, in the ADS testing task, the reconstruction should also keep the key information, especially the location of objects. To ensure the representation could be effectively used for ADS, the problem we address is to ensure **perception stability**: the outputs of the perception model $\mathcal{P}(\cdot)$ applied to the reconstructed model should remain consistent with those obtained from the ground truth scene, defined as

$$\mathcal{L}_{\text{perc}} = d_{\text{perc}}(\mathcal{P}(\mathcal{R}(x)), \mathcal{P}(x)) \tag{2}$$

where $d_{perc}(\cdot, \cdot)$ measures the discrepancy at the semantic or perception level.

Formally, we formulate the problem as a constrained optimization: the reconstruction quality must be sufficiently preserved, while minimizing the perception discrepancy.

$$\begin{aligned} \min_{\mathcal{R}} \quad & \mathbb{E}_x \big[ d_{\text{perc}}\big(\mathcal{P}(\mathcal{R}(x)), \mathcal{P}(x)\big) \big] \\ \text{s.t.} \quad & \mathbb{E}_x \big[ d_{\text{img}}\big(\mathcal{R}(x), x\big) \big] \leq \varepsilon, \end{aligned} \tag{3}$$

## 3.2 PRELIMINARY STUDY

Based on the problem definition, we would like first to explore the limitations of existing 3DGS methods in the traffic scene reconstruction task.

### 3.2.1 EXPERIMENTS SETUP

**3DGS approaches** We select three state-of-the-art 3DGS approaches: $S^3$Gaussian (Huang et al., 2024), OmniRe (Chen et al., 2025), and EMD (Wei et al., 2025). In practice, EMD is built upon either $S^3$ Gaussian or OmniRe as its base, and we conduct separate experiments for each base. Furthermore, all reconstruction training processes consist of 5,000 steps in the coarse stage, followed by 30,000 steps in the fine stage.

**Perception Model** To avoid the influence of the perception model's architecture, we evaluate perception performance using three popular detection models with different architectures: YOLOv8 (Varghese & M., 2024), Faster R-CNN (Ren et al., 2015) and RT-DETR(Zhao et al., 2024).

**Metrics** As described, we aim to investigate the performance of reconstruction methods in terms of both visual quality and perception stability, and we use metrics from both perspectives for comparison accordingly. We utilize *SSIM* to characterize and measure visual quality. Meanwhile, to maintain perception stability, we use *mean IoU* to measure detection differences, and *mAP@*[0.5:0.95] to quantify detection confidence and mis-classification. Additionally, we counted the number of missed detections after reconstruction.

**Ground truth** It should be emphasized that the calculation of the perception stability part does not directly compare against the object information in the ground truth; instead, it is based on the outputs of the same perception model when fed with the ground truth.

**Experiment scenes** We fully reproduced all experimental scenes of $S^3$Gaussian and OmniRe. For the EMD method, experiments were conducted using the scenes corresponding to each respective base method. To limit space, all results reported in the paper are averages across all scenes.

### 3.2.2 PRILMINARY RESULTS

Table 1 shows that existing 3DGS methods generally achieve high visual quality. It can be observed that existing methods already exhibit high performance in visual metrics —whether in SSIM, PSNR, or LPIPS- but their perception stability is not satisfactory, yet they are fair to use.

To better understand this mismatch, we further examined the statistical relationship between pixel-level metrics and perception stability, shown as table 2. Although these visual metrics do show

Table 1: Visual Quality and Perception Metrics of Existing Methods (On Average)

| | pixel Level | | | YOLO v8 | | | Faster RCNN | | |
|---|---|---|---|---|---|---|---|---|---|
| | SSIM | PSNR | LPIPS | mAP | mean IOU | Miss | mAP | mean IOU | Miss |
| $S^3$Gaussian | 0.924 | 31.27 | 0.105 | 0.550 | 0.803 | 1.5 | 0.171 | 0.620 | 2.0 |
| OmniRe | 0.953 | 33.77 | 0.049 | 0.489 | 0.832 | 0.0 | 0.320 | 0.718 | 0.3 |
| EMD($S^3$G) | 0.923 | 31.39 | 0.104 | 0.578 | 0.755 | 0.0 | 0.270 | 0.689 | 0.5 |
| EMD(OmniRe) | 0.962 | 35.02 | 0.039 | 0.452 | 0.839 | 0.3 | 0.348 | 0.768 | 0.3 |

Table 2: The statistical correlation between pixel-level metrics(mAP) and detection stability with Yolo v8. Limited to page size, more data(correlations with meanIOU) will be shown in the Appendix. $r$ denotes the Pearson correlation coefficient, and $p$ denotes the $p$-value.

| Correlation | $r_{\text{SSIM}}$ | $p_{\text{SSIM}}$ | $r_{\text{PSNR}}$ | $p_{\text{PSNR}}$ | $r_{\text{LPIPS}}$ | $p_{\text{LPIPS}}$ |
|---|---|---|---|---|---|---|
| $S^3$Gaussian | 0.767 | 2.43E-3 | 0.658 | 3.09E-3 | -0.721 | 1.21E-3 |
| OmniRe | 0.417 | 3.11E-3 | 0.484 | 1.07E-3 | -0.568 | 2.50E-3 |
| EMD($S^3$G) | 0.374 | 2.14E-3 | 0.319 | 1.09E-3 | -0.387 | 2.33E-3 |
| EMD(OmniRe) | 0.295 | 1.41E-3 | 0.439 | 1.52E-4 | -0.369 | 3.76E-3 |

statistically significant correlations with mAP and mean IoU (all $p < 5E-3$), their Pearson coefficients remain only in the range of 0.3–0.6. This indicates that the connection between the two is fairly weak: visual improvements tend to move in the same direction as perception stability, but the effect is limited and far from predictive.

## 4 METHOD 1: PERCEPTION-ALIGNED LOSS INTEGRATION

As we found in Section 3.2, in the ADS traffic scene generation task, existing 3DGS methods could achieve high visual quality, but often fail to guarantee stable perception outputs for objects. To address this limitation, we first propose a naive yet intuitive solution that integrates detection results as an additional loss term directly into the 3DGS model training process. This approach would encourage the training process not only to focus on visual quality but also to preserve stable perception results.

Formally, we define the perception-aligned loss $\mathcal{L}_{\text{perc}}$ as the sum of the errors in the predicted bounding boxes and their associated class labels as

$$\mathcal{L}_{\text{perc}} = \sum_i \left( \lambda_{\text{box}} \cdot \mathcal{L}_{\text{box}}(\mathcal{B}(x), \mathcal{B}(\mathcal{R}(x))) + \lambda_{cls} \cdot \mathcal{L}_{\text{cls}}(\mathcal{C}(x), \mathcal{C}(\mathcal{R}(x))) \right) \tag{4}$$

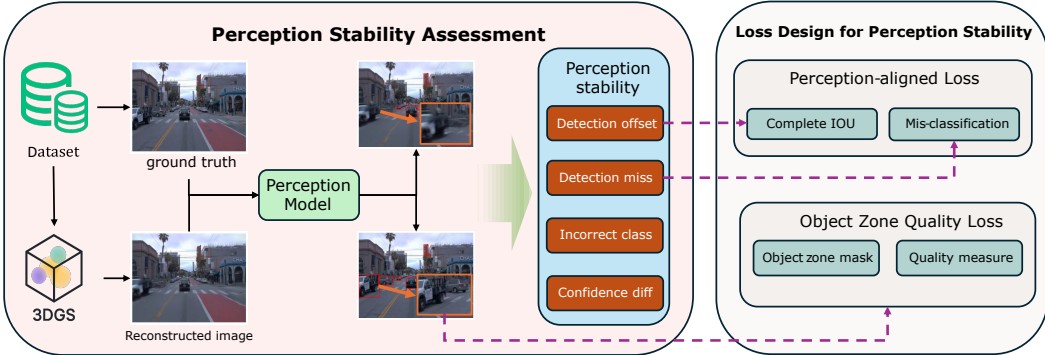

Figure 2: Overview of this work. Perception stability is measured by comparing the outputs of the same perception model when fed with the original frames versus the reconstructed frames. Based on the perception outputs and the object regions identified by the perception model, we designed a perception-aligned loss and an object zone quality loss to improve perception stability.

where $\mathcal{B}, \mathcal{C} \in \mathcal{P}$ refers to the bounding box and classification label in the perception model results, respectively, and $\lambda$ is the weight of corresponding items.

In autonomous driving tasks, object bounding boxes not only require sufficient overlap with the ground truth but also accurate center positions, as these factors greatly affect subsequent modules such as tracking and prediction (Yin et al., 2021; Sharath & Mehran, 2021). Meanwhile, the aspect ratio of the bounding box also affects the ADS's classification of the target type and can influence subsequent decision-making (Luo et al., 2021). **CIoU**(Complete IoU) (Zheng et al., 2021) naturally satisfies both of these requirements by jointly penalizing deviations in overlap, center-point distance, and aspect ratio, making it particularly suitable as a loss function for bounding box regression in ADS scenarios. And the perception-aligned loss is defined as

$$\mathcal{L}_{\text{box}} = 1 - \frac{1}{n} \sum_{i=1}^{n} \text{CIoU}\big(\mathcal{B}_i(x), \mathcal{B}_i(\mathcal{R}(x))\big) \tag{5}$$

where $\mathcal{B}_i$ is the the bounding box of the $i$-th object in a given frame and $n$ is the total number of objects in the frame.

On the other hand, the classification loss is computed directly based on whether the predicted class labels are the same in the ground truth frame and the reconstruction frame, as

$$\mathcal{L}_{\text{cls}} = 1 - \frac{1}{n} \sum_{i=1}^{n} \mathbf{1}\big(\mathcal{C}_i(x) = \mathcal{C}_i(\mathcal{R}(x))\big) \tag{6}$$

Finally, this perception-aligned loss is integrated into the total training objective

$$\mathcal{L}_{total} = \lambda_{\text{visual}} \cdot \mathcal{L}_{\text{visual}} + \lambda_{\text{perc}} \cdot \mathcal{L}_{\text{perc}} \tag{7}$$

where $\mathcal{L}_{\text{visual}}$ is the existing visual quality-aware loss function. Note that this loss is only backpropagated during 3DGS training, while the perception model remains frozen and is not updated in any way.

## 5 Experiments for Method 1

### 5.1 Experiments Design

To evaluate the effect of integrating perception-aware losses into the 3DGS training, we use YOLOv8 as the perception model. We first apply YOLOv8 to the frames of the training data to obtain $\mathcal{P}(x)$. During the 3DGS training process, the same YOLOv8 is then applied to each reconstructed frame to extract $\mathcal{P}(\mathcal{R}(x))$. The perception-aligned loss is computed according to equations 4 and 6 and integrated into the 3DGS training objective in equation 7, guiding the reconstruction to preserve stable perception outputs. Similar to existing work, this loss will be applied only in the fine-stage training.

To simplify the choice of parameters and to focus on the effect of the perception-aligned loss, we set all $\lambda$ values ($\lambda_{\text{box}}, \lambda_{\text{cls}}, \lambda_{\text{visual}}, \lambda_{\text{perc}}$) to 1. This allows us to clearly see how the perception-aligned loss affects the reconstruction, without the results being influenced by different weighting factors.

All comparison methods, scenes, and metric selections are kept the same as in Section 3.2.1.

### 5.2 Experiment Results

Table 3 shows the results of perception-aligned loss integrated into the training process. We can draw the following conclusions: *1)* Integrating the perception-aligned loss leads to a significant improvement in mAP and mean IoU, indicating that the reconstructed scenes achieve stronger perception stability; *2)* The same trend is observed on Faster R-CNN and RT-DETR, in addition to the YOLOv8 used during training, suggesting that the improved perception stability is not limited to the training model but indeed reflects an enhancement in reconstruction quality; *3)* The visual quality loss exhibits only minor fluctuations ($\pm < 1\%$), indicating that adding the perception-aligned loss has a relatively small and acceptable impact on visual quality.

Table 3: Perception-aligned Loss Integration, use YOLOv8 as guidance model, test Faster RCNN and RT-DETR as black-box model

| | Pixel-Level | | | YOLOv8 | | |
|---|---|---|---|---|---|---|
| | SSIM↑ | PSNR↑ | LPIPS↓ | mAP↑ | mean IoU↑ | Miss↓ |
| $S^3$Gaussian | 0.924 | 31.27 | 0.106 | 0.550 | 0.803 | 1.5 |
| $S^3$Gaussian+$\mathcal{L}_{\text{perc}}$ | 0.920 | 31.53 | 0.106 | 0.593 | 0.840 | 0.83 |
| OmniRe | 0.953 | 33.77 | 0.049 | 0.489 | 0.832 | 0.0 |
| OmniRe+$\mathcal{L}_{\text{perc}}$ | 0.954 | 33.75 | 0.048 | 0.507 | 0.845 | 0.0 |
| EMD($S^3$G) | 0.923 | 32.89 | 0.057 | 0.578 | 0.755 | 0.0 |
| EMD($S^3$G)+$\mathcal{L}_{\text{perc}}$ | 0.951 | 33.37 | 0.046 | 0.583 | 0.857 | 0.0 |
| EMD(OmniRe) | 0.962 | 35.02 | 0.039 | 0.452 | 0.839 | 0.3 |
| EMD(OmniRe)+$\mathcal{L}_{\text{perc}}$ | 0.954 | 35.42 | 0.035 | 0.497 | 0.846 | 0.0 |

| | Faster R-CNN | | | RT-DETR | | |
|---|---|---|---|---|---|---|
| | mAP↑ | mean IoU↑ | Miss↓ | mAP↑ | mean IoU↑ | Miss↓ |
| $S^3$Gaussian | 0.171 | 0.620 | 2.0 | 0.494 | 0.829 | 0.0 |
| $S^3$Gaussian+$\mathcal{L}_{\text{perc}}$ | 0.229 | 0.632 | 0.7 | 0.509 | 0.829 | 0.0 |
| OmniRe | 0.320 | 0.718 | 0.3 | 0.519 | 0.789 | 0.0 |
| OmniRe+$\mathcal{L}_{\text{perc}}$ | 0.360 | 0.722 | 0.0 | 0.526 | 0.792 | 0.0 |
| EMD($S^3$G) | 0.270 | 0.689 | 0.5 | 0.518 | 0.770 | 0.0 |
| EMD($S^3$G)+$\mathcal{L}_{\text{perc}}$ | 0.307 | 0.701 | 0.0 | 0.674 | 0.875 | 0.0 |
| EMD(OmniRe) | 0.348 | 0.768 | 0.3 | 0.542 | 0.852 | 0.0 |
| EMD(OmniRe)+$\mathcal{L}_{\text{perc}}$ | 0.352 | 0.779 | 0.0 | 0.542 | 0.842 | 0.0 |

# 6 METHOD 2: OBJECT ZONE QUALITY LOSS FOR RECONSTRUCTION

## 6.1 REVIEW OF PERCEPTION-ALIGNED LOSS INTEGRATION

In the experiments presented in Section 5, we have demonstrated the positive effect of the perception-aligned loss on perception stability. However, this approach still has two limitations: *1)* the improvement in perception is derived from the perception model's outputs, which limits interpretability; *2)* during training, inference through the perception model is required at every iteration, significantly increasing training time and memory consumption. Therefore, we proceed to further analysis based on the existing results.

Through a study of numerous cases where the perception model is affected after reconstruction, we observed two typical phenomena, as shown in Figure 3. The first, illustrated in I, shows discontinuities in object regions, which mostly occur in static objects. The second, shown in II, exhibits blurring in object regions, and this is most pronounced in dynamic objects. Both issues point to a common problem: current 3DGS methods for traffic scene reconstruction tend to produce a larger shortfall in object regions compared to other layers, such as the sky or buildings. Previous studies have shown that, due to its reliance on LiDAR or other point cloud data, 3DGS often lacks fine-grained reconstruction at object edges (Chelani et al., 2025; Lu et al., 2025), resulting in blurred object edges and perceptual changes. Furthermore, in most scenes, object regions occupy a much smaller area than other layers. Under existing methods, the reconstruction quality of object regions is even less reliably guaranteed.

## 6.2 OBJECT ZONE QUALITY LOSS

Based on the analysis in Section 6.1, we believe that a core reason for the instability of perception after reconstruction is the relatively low reconstruction quality of object regions. Thus, we propose

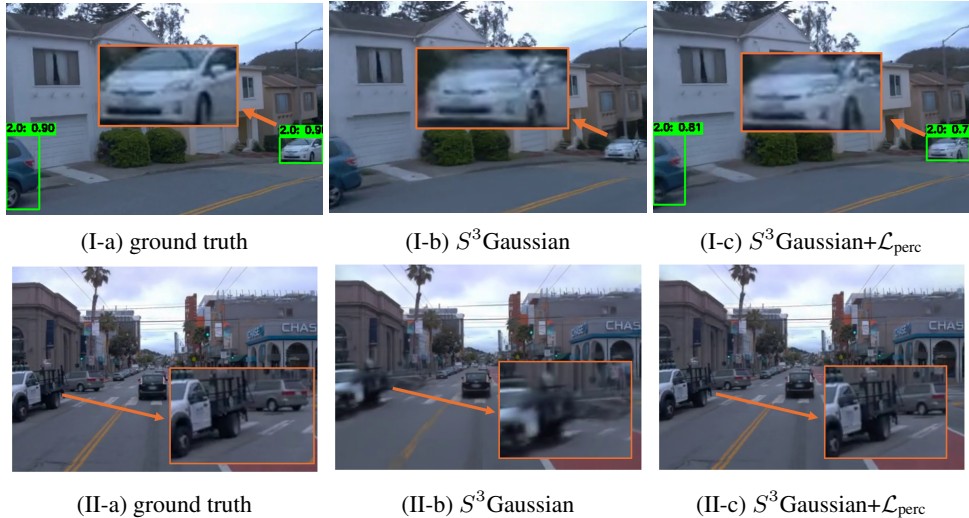

(I-a) ground truth        (I-b) $S^3$Gaussian        (I-c) $S^3$Gaussian$+\mathcal{L}_{\text{perc}}$

(II-a) ground truth        (II-b) $S^3$Gaussian        (II-c) $S^3$Gaussian$+\mathcal{L}_{\text{perc}}$

Figure 3: Comparison of reconstructed for $S^3$Gaussian, the perception model fails to maintain original outputs, whereas integrating the perception-aligned loss leads to improved perception consistency. I: modelling fractures; II: object zone blur

a more intuitive solution: we design a custom **object zone quality loss** that computes a separate visual quality loss for the object regions identified by the perception model.

$$\mathcal{L}_{\text{obj-vis}} = d_{\text{vis}}\big(\mathcal{R}(x) \odot \mathcal{B}(x),\ x \odot \mathcal{B}(x)\big) \tag{8}$$

where $\odot$ applies the mask to extract the object zones, and $d_{\text{vis}}$ is a visual similarity metric. Then, similar to perception-aligned loss integration, the object zone quality loss is utilized as

$$\mathcal{L}_{total} = \lambda_{\text{visual}} \cdot \mathcal{L}_{\text{visual}} + \lambda_{\text{obj-vis}} \cdot \mathcal{L}_{\text{obj-vis}} \tag{9}$$

By introducing the proposed object-zone quality loss, the reconstruction process is clearly guided to emphasize regions that the perception model deems critical in the ground-truth frames (i.e., the detection zone). This targeted enhancement improves the fidelity of object regions, thereby improving perception stability. Furthermore, since the training procedure relies solely on offline ground-truth perception results, this approach offers improved computational efficiency compared to Method 1, which requires direct incorporation of perception-aligned loss for each iteration.

# 7 EXPERIMENTS FOR METHOD 2

## 7.1 EXPERIMENT DESIGN

When evaluating reconstruction quality, this experiment introduces a new metric, the object zone loss, which measures reconstruction quality specifically within the object regions identified by the perception model in the ground-truth images. In addition, as mentioned in Section 6.2, the object zone quality loss reduces computational complexity compared to the perception-aligned loss. We also record and compare the time cost during the reconstruction training process to quantify this efficiency difference.

## 7.2 EXPERIMENT RESULTS

Table 4 shows the results of object zone quality loss integrated into the training process. The experimental results indicate that:

Table 4: Object Zone Quality Loss Integration, use YOLOv8 as guidance model, test Faster RCNN and RT-DETR as black-box model

| | Pixel-Level | | | | YOLO v8 | | |
| --- | --- | --- | --- | --- | --- | --- | --- |
| | SSIM↑ | Obj SSIM↑ | PSNR↑ | LPIPS↓ | mAP↑ | mean IOU↑ | Miss↓ |
| $S^3$Gaussian | 0.924 | 0.877 | 31.27 | 0.106 | 0.550 | 0.803 | 1.5 |
| $S^3$Gaussian+$\mathcal{L}_{\text{perc}}$ | 0.920 | 0.897 | 31.53 | 0.106 | 0.593 | 0.840 | 0.83 |
| $S^3$Gaussian+$\mathcal{L}_{\text{obj-vis}}$ | 0.937 | 0.921 | 31.89 | 0.082 | 0.672 | 0.862 | 0.4 |
| $S^3$Gaussian+$\mathcal{L}_{\text{perc}}$+$\mathcal{L}_{\text{obj-vis}}$ | **0.941** | **0.924** | **32.00** | **0.083** | **0.700** | **0.872** | **0.0** |
| OmniRe | 0.953 | 0.867 | 33.77 | 0.049 | 0.489 | 0.832 | 0.0 |
| OmniRe+$\mathcal{L}_{\text{perc}}$ | 0.954 | 0.876 | 33.75 | 0.048 | 0.507 | 0.845 | 0.0 |
| OmniRe+$\mathcal{L}_{\text{obj-vis}}$ | 0.949 | 0.893 | 33.25 | 0.046 | 0.545 | 0.856 | 0.0 |
| OmniRe+$\mathcal{L}_{\text{perc}}$+$\mathcal{L}_{\text{obj-vis}}$ | **0.957** | **0.899** | **33.75** | **0.047** | **0.609** | **0.870** | **0.0** |
| EMD($S^3$G) | 0.923 | 0.855 | 32.89 | 0.057 | 0.578 | 0.755 | 0.0 |
| EMD($S^3$G)+$\mathcal{L}_{\text{perc}}$ | 0.951 | 0.923 | 33.37 | 0.046 | 0.583 | 0.857 | 0.0 |
| EMD($S^3$G)+$\mathcal{L}_{\text{obj-vis}}$ | 0.952 | 0.946 | 33.56 | 0.043 | 0.601 | 0.843 | 0.0 |
| EMD($S^3$G)+$\mathcal{L}_{\text{perc}}$+$\mathcal{L}_{\text{obj-vis}}$ | **0.952** | **0.948** | **33.49** | **0.043** | **0.600** | **0.860** | **0.0** |
| EMD(OmniRe) | 0.962 | 0.910 | 35.02 | 0.039 | 0.452 | 0.839 | 0.3 |
| EMD(OmniRe)+$\mathcal{L}_{\text{perc}}$ | 0.954 | 0.915 | 35.42 | 0.035 | 0.497 | 0.846 | 0.0 |
| EMD(OmniRe)+$\mathcal{L}_{\text{obj-vis}}$ | 0.965 | 0.942 | 35.20 | 0.034 | 0.508 | 0.856 | 0.0 |
| EMD(OmniRe)+$\mathcal{L}_{\text{perc}}$+$\mathcal{L}_{\text{obj-vis}}$ | **0.969** | **0.940** | **35.33** | **0.035** | **0.510** | **0.856** | **0.0** |

| | Faster RCNN | | | RT-DETR | | |
| --- | --- | --- | --- | --- | --- | --- |
| | mAP↑ | mean IOU↑ | Miss↓ | mAP↑ | mean IOU↑ | Miss↓ |
| $S^3$Gaussian | 0.171 | 0.620 | 2.0 | 0.494 | 0.829 | 0.0 |
| $S^3$Gaussian+$\mathcal{L}_{\text{perc}}$ | 0.229 | 0.632 | 0.7 | 0.509 | 0.829 | 0.0 |
| $S^3$Gaussian+$\mathcal{L}_{\text{obj-vis}}$ | 0.271 | 0.689 | 0.4 | 0.603 | 0.843 | 0.0 |
| $S^3$Gaussian++$\mathcal{L}_{\text{perc}}$+$\mathcal{L}_{\text{obj-vis}}$ | **0.269** | **0.695** | **0.3** | **0.610** | **0.845** | **0.0** |
| OmniRe | 0.320 | 0.718 | 0.3 | 0.519 | 0.789 | 0.0 |
| OmniRe+$\mathcal{L}_{\text{perc}}$ | 0.360 | 0.722 | 0.0 | 0.526 | 0.792 | 0.0 |
| OmniRe+$\mathcal{L}_{\text{obj-vis}}$ | 0.343 | 0.735 | 0.0 | 0.524 | 0.800 | 0.0 |
| OmniRe+$\mathcal{L}_{\text{perc}}$+$\mathcal{L}_{\text{obj-vis}}$ | **0.359** | **0.740** | **0.0** | **0.675** | **0.874** | **0.0** |
| EMD($S^3$G) | 0.270 | 0.689 | 0.5 | 0.518 | 0.770 | 0.0 |
| EMD($S^3$G)+$\mathcal{L}_{\text{perc}}$ | 0.308 | 0.715 | 0.1 | 0.674 | 0.875 | 0.0 |
| EMD($S^3$G)+$\mathcal{L}_{\text{obj-vis}}$ | 0.308 | 0.701 | 0.0 | 0.666 | 0.870 | 0.0 |
| EMD($S^3$G)+$\mathcal{L}_{\text{perc}}$+$\mathcal{L}_{\text{obj-vis}}$ | **0.311** | **0.717** | **0.0** | **0.677** | **0.879** | **0.0** |
| EMD(OmniRe) | 0.348 | 0.768 | 0.3 | 0.542 | 0.852 | 0.0 |
| EMD(OmniRe)+$\mathcal{L}_{\text{perc}}$ | 0.352 | 0.779 | 0.0 | 0.542 | 0.842 | 0.0 |
| EMD(OmniRe)+$\mathcal{L}_{\text{obj-vis}}$ | 0.389 | 0.778 | 0.0 | 0.558 | 0.862 | 0.0 |
| EMD(OmniRe)+$\mathcal{L}_{\text{perc}}$+$\mathcal{L}_{\text{obj-vis}}$ | **0.404** | **0.783** | **0.0** | **0.562** | **0.879** | **0.0** |

- The application of the object zone quality loss explicitly emphasizes these regions as a training focus, leading to a significant improvement in the reconstruction quality of the targeted areas. At the same time, this also has a positive effect on the overall visual quality.
- The perception stability of the reconstructed scenes has been significantly improved, which also implies an enhancement in the reconstruction quality of object regions, with more detailed information such as edges and textures being preserved. This, in turn, benefits the recognition performance of perception models.
- Compared to the naive approach that directly employs a perception-aligned loss, the object zone quality loss provides more stable improvements in reconstruction quality. However, in specific scenarios, the IoU accuracy does not necessarily outperform that of the perception-aligned loss.
- Optimizing with both losses simultaneously can, in most cases, yield better results compared to using either loss individually, although the improvement is relatively modest.

## 7.3 RUNTIME ANALYSIS

Table 5: Average reconstruction training process time consumption.

|  | per 100 epochs(seconds) | | | in total (minutes) | | |
|---|---|---|---|---|---|---|
|  | origin | with $\mathcal{L}_{\text{perc}}$ | with $\mathcal{L}_{\text{obj-vis}}$ | origin | with $\mathcal{L}_{\text{perc}}$ | with $\mathcal{L}_{\text{obj-vis}}$ |
| S3G | 25.20 | 26.67 | 25.30 | 204.4 | 232.2 | 205.4 |
| OmniRe | 34.84 | 36.07 | 34.91 | 282.1 | 332.2 | 283.5 |
| EMD(S3G) | 32.44 | 33.94 | 32.51 | 262.9 | 312.9 | 263.9 |
| EMD(OmniRe) | 44.94 | 46.45 | 45.00 | 364.5 | 413.2 | 364.9 |

As shown in the Figure 5, incorporating perception loss into the training process requires an additional YOLO inference at each iteration, leading to a noticeable increase in training time. In contrast, the use of object zone quality loss, although introducing an extra loss computation, has only a negligible impact on overall runtime. Note that: *1)* these results are obtained on our experimental platform with an RTX A5000 GPU, and the absolute values may vary depending on the computing device; *2)* In our experiments, we employ the lightweight YOLOv8n model; using a heavier detector, such as Faster R-CNN, would likely exacerbate this difference.

## 8 DISCUSSION

**Loss Weight and Trade-offs**: In our experiments, all loss weights ($\lambda$) were set to 1 to demonstrate the effectiveness of the proposed methods to avoid the effect of parameter selection. However, this work does not fully explore the trade-off between maintaining high visual realism and ensuring stable perception outputs by adjusting the loss weights. For example, in our task, the scenes contain many objects, but each occupies a relatively small area. In contrast, robotic manipulation tasks often involve key targets within the robot's field of view that occupy a larger portion of the scene, but are typically few in number. Clearly, these two scenarios would require different combinations of $\lambda$ values, a topic not explored in this work. Future work could study adaptive or learned weighting strategies to balance this trade-off more effectively.

**Broader Applicability:** While this work focuses on traffic scene reconstruction for autonomous driving systems, the principle of perception-aware reconstruction may not be limited to this domain. Any modular AI system that relies on perception outputs for downstream tasks—such as robotics, AR/VR, or automated inspection systems—could potentially benefit from our approach. For example, in robotic manipulation, an unstable perception of object positions could result in incorrect grasps or collisions; in AR/VR, inconsistent perception could disrupt object alignment and user experience; in automated inspection systems, unstable detection could lead to missed defects. Our approach enhances reliability and trustworthiness in reconstructed scenes across these domains. From the current perspective, our method does not rely on strong domain-specific knowledge. In principle, it could be easily adapted to these perception-critical tasks.

## 9 CONCLUSION

We introduced **perception-aware reconstruction**, aiming to ensure 3DGS preserve both visual quality and perception stability in ADS traffic scene reconstruction. Through our experiments, we found that existing methods, while improving visual quality, do not guarantee an increase in perception stability. To address this, we propose two approaches—perception-aligned loss and object zone quality loss—to effectively improve perception stability across multiple 3DGS methods and detection models. Our results demonstrate that both approaches can significantly enhance the perception stability of reconstructed scenes. This work points the way toward 3D reconstructions that are both realistic and practical for safety-critical applications.

## 10 Acknowledgement

This work was supported in part by JST CRONOS Grant (No. JPMJCS24K8), JSPS KAKENHI Grant (No.JP21H04877, No.JP23H03372, and No.JP24K02920), Canada CIFAR AI Chairs Program, the Natural Sciences and Engineering Research Council of Canada.

## A Reproducibility Statement

### A.1 Reconstructed Scenes

Our experiments are entirely based on the Waymo dataset, using the same scene selection as in $S^3$Gaussian and EMD.

For the experiments of $S^3$Gaussian and EMD($S^3$Gaussian), we use scene ids: 003, 019, 021, 022, 036, 069, 081, 094, 126, 139, 140, 146, 148, 157, 181, 200, 204, 237, 241, 297, 302, 314, 362, 427, 482, 495, 524, 527, 581, 700, 753, 780, 795

For the experiments of OmniRe and EMD(OmniRe), we use scene ids: 014, 016, 021, 022, 023, 031, 049, 053, 064, 080, 088, 094, 111, 114, 222, 327, 552, 621, 700, 784, 785, 788, 796

### A.2 Experimental Environment

We conduct our experiments primarily on a server with an Intel 10920X and dual NVIDIA RTX A5000 GPUs. In practice, most experiments can be completed with 24GB of VRAM. Few scenes (like 788) would take more VRAM, and we deploy these scene experiments on a server with an NVIDIA V100 with 32 GB of VRAM.

### A.3 Implementation Details

Here are some implementation details not mentioned in the main text.

**Perception model weights**

YOLO: yolov8n from ultralytics;

Faster RCNN: fasterrcnn_resnet50_fpn from torchvision

**Image load size** The image load size of the four base approaches is not the same, and we set it to [320, 480] for each camera.

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
