# OpenReview forum: "Beyond Visual Reconstruction Quality: Object Perception-aware 3D Gaussian Splatting for Autonomous Driving"
_ICLR.cc/2026/Conference — ICLR 2026 Poster_

### Official Review · Reviewer_7ST4 · 2025-10-15

**Soundness:** 3
**Presentation:** 3
**Contribution:** 2
**Rating:** 2
**Confidence:** 4

**Summary:**

This paper identifies a gap in 3D Gaussian Splatting for autonomous driving: existing methods optimize global visual similarity, but this does not ensure consistent perception outputs, which are crucial for ADS behavior. It proposes perception-aware reconstruction with two losses: a perception-aligned loss penalizing output inconsistencies, and an object-zone quality loss emphasizing task-relevant regions. Experiments show improved perception stability while maintaining visual fidelity.

**Strengths:**

This paper presents novel evaluation metrics that specifically assess perception performance, which is meaningful for autonomous driving simulation.

**Weaknesses:**

1. One weakness is that the results appear to be highly dependent on the chosen perception model. While testing with Faster R-CNN, more baseline methods should be included for comparison.

2. As shown in Table 2, the Perception-Aligned Loss leads to a drop in SSIM for some methods. While the idea of using perception scores is meaningful, the trade-off between visual quality and perception alignment is not clearly analyzed.

3. The Object Zone metric is fairly straightforward and not novel, as similar concepts such as Dynamic Region PSNR have been used in prior works.

**Questions:**

To clearly demonstrate the effect of each component, a combination of Object Zone Quality Loss and Perception-Aligned Loss integration should be evaluated in Table 3.

---

> ### Author Response · Authors · 2025-11-19
>
> Thank you for reviewing our paper and providing valuable feedback. We have carefully analyzed the issues you raised and will respond in detail below. We believe that after clarifying some misunderstandings and supplementing the necessary experiments, you will see the novelty and contribution of our work.
>
> Weakness 1: One weakness is that the results appear to be highly dependent on the chosen perception model. While testing with Faster R-CNN, more baseline methods should be included for comparison.
>
> Ans： Thank you for your suggestion. We conducted supplementary experiments using RT-DETR. The results, guided by information provided by YOLO during the training phase, were evaluated using RT-DETR. Taking EMD ($S^3G$) as an example, the results are as follows:
>
> & mAP   & meanIOU & map   & meanIOU & Miss \\
>
> EMD($S^3G$) & 0.923 & 0.855   & 0.518 & 0.770   & 0    \\
>
> EMD($S^3G$)+ $\mathcal{L}_\text{perc}$       & 0.951 & 0.923   & 0.674 & 0.875   & 0    \\
>
>  EMD($S^3G$)+ $\mathcal{L}_\text{obj-vis}$           & 0.952 & 0.946   & 0.666 & 0.870   & 0
>
> More results will be provided in the revised PDF later. The data shows that the conclusion remains unchanged after using the DETR architecture detector: our method can achieve higher perception stability without reducing pixel-level performance, and this perception stability is not dependent on specific detector.
>
> Weakness 2: As shown in Table 2, the Perception-Aligned Loss leads to a drop in SSIM for some methods. While the idea of using perception scores is meaningful, the trade-off between visual quality and perception alignment is not clearly analyzed.
>
> The slight decrease of -0.008 in OmniRe shown in Table 2 is indeed present. In this study, we consider it a beneficial trade-off rather than a fatal flaw.
>
> Our core argument is that for autonomous driving testing tasks, 3DGS needs to focus on perception stability to ensure that the generated scene is meaningful for testing. In this case, the trade-off of a 0.8% decrease in SSIM for a 4.2% improvement in mAP and a 0.7% improvement in meanIoU is considered very worthwhile. Furthermore, in most cases, pixel-level quality did not decrease and even improved, indicating that our method can generally improve or at least maintain visual quality simultaneously, while the small compromises made in individual cases yield significant perceptual gains.
>
>
> Weakness 3: The Object Zone metric is fairly straightforward and not novel, as similar concepts such as Dynamic Region PSNR have been used in prior works.
>
> We respectfully disagree with this assessment. Our main contribution is not introducing a new evaluation metric. Instead, we are the first to bring the concept of “perception stability” into 3D reconstruction tasks and design a specific training objective to achieve it.
>
> This is a paradigm shift: in prior work, such as Dynamic Region PSNR, still focuses on improving visual quality. In contrast, our L_{obj-vis} directly optimizes the consistency of outputs from downstream perceptual models—something no previous method has done. This creates a direct path from reconstruction to perception, which is the core novelty of our work.
>
> \
> \
>
> Question 1: To clearly demonstrate the effect of each component, a combination of Object Zone Quality Loss and Perception-Aligned Loss integration should be evaluated in Table 3.
>
> We immediately began deploying this experiment, and have already completed the experiment with the $S^3$Gaussian group and EMD($S^3$G) group. Here is a comparison with the L_{obj-vis} group, which has the best overall performance.
>
> (YOLOv8) & SSIM$\uparrow$ & Obj SSIM$\uparrow$ & mAP$\uparrow$           & mean IOU$\uparrow$   &  Miss$\downarrow$  (FasterRCNN)   & mAP $\uparrow$        & mean IOU$\uparrow$ &  Miss$\downarrow$ \\
>
> $S^3$Gaussian+$\mathcal{L}_\text{obj-vis}$ & 0.949 & 0.893 & 0.545 & 0.856 & 0.0 & 0.343 & 0.73 & 0.0\
>
> $S^3$Gaussian +$L_\text{perc}$ + $\mathcal{L}_\text{obj-vis}$& 0.939 & 0.923 &  0.682 &0.888& 0.4 & 0.319 & 0.730 & 0.0\
>
> EMD($S^3$G)+$\mathcal{L}_\text{obj-vis}$ & 0.952 & 0.946 & 0.601 &0.843 & 0.0 & 0.308 &  0.701 & 0.0}\\
>
> EMD($S^3$G) +$L_\text{perc}$+$\mathcal{L}_\text{obj-vis}$ & 0.954 & 0.951 & 0.619 &0.880 & 0.0 & 0.318 &  0.722 & 0.0}\\
>
> More experiments will be updated in the revisde PDF version after the run is complete. The results show that a combination of Object Zone Quality Loss and Perception-Aligned Loss integration does indeed achieve better results than applying only one optimization method.

---

### Official Review · Reviewer_sxkk · 2025-10-27

**Soundness:** 3
**Presentation:** 4
**Contribution:** 4
**Rating:** 6
**Confidence:** 5

**Summary:**

This paper points out that existing 3D Gaussian Splatting (3DGS) methods for street-scene reconstruction overlook the essential requirement of preserving autonomous driving system (ADS) behavior. The authors argue that optimizing solely for visual reconstruction performance fails to ensure consistency in downstream perception modules. To enhance the reconstruction quality of object instances, the paper introduces two complementary losses: Perception-Aligned Loss and Object-Zone Quality Loss. Experimental results demonstrate the effectiveness of the proposed approach in improving perception consistency without compromising visual fidelity.

**Strengths:**

1. The paper is well-motivated, making its ideas easy to follow.
2. Simple yet effective methods: perception-aligned and object-zone losses are easy to implement and show consistent gains.
3. Comprehensive evaluation: multiple base 3DGS models, perception backbones, and visual/perceptual metrics are tested.

**Weaknesses:**

1. The writing and figures need to be refined, especially in the Introduction and Method section. The figures contain text that is too small to read clearly and would benefit from clearer visual design and labeling.
2. The Broader Applicability discussion provides limited new insights. The paper would be stronger if it included a deeper analysis or ablation of how the proposed methods generalize across different tasks or domains.
3. The performance improvement appears somewhat incremental. Evaluating the method on additional and more diverse scenarios could better demonstrate its effectiveness and practical significance.

**Questions:**

1. Would an adaptive λ scheduling strategy (balancing Lvisual vs Lperc/Lobj) further improve trade-offs?

---

> ### Author Response · Authors · 2025-11-19
>
> We sincerely thank you for your high praise of our work. Your suggestions for improvement are very pertinent, and we will do our best to implement them in the revised version to improve the overall quality of the paper.
>
> Weakness 1: The writing and figures need to be refined, especially in the Introduction and Method section. The figures contain text that is too small to read clearly and would benefit from clearer visual design and labeling.
>
> Ans: Thank you very much for reminding us. We have carefully reviewed the original text again and indeed found some issues that we hadn't noticed during submission (such as the inconsistency between obj-vis and vis-obj in table3). We will also recreate the figures to improve the reading experience.
>
> Weakness 2: The Broader Applicability discussion provides limited new insights. The paper would be stronger if it included a deeper analysis or ablation of how the proposed methods generalize across different tasks or domains.
>
> Ans: Yes, our research method is not only applicable to the perception model; the same methodology can be used for many other modules. Regarding the downstream work in this paper, for the prediction module in ADS, the perception output serves as the prediction input. Theoretically, higher perception quality should lead to better prediction stability, but this is not always the case. We can identify the correlation between perception stability and prediction stability. If a gap exists, we can also propose targeted supplementary solutions based on the characteristics of the prediction module. The same logic applies to planning and control. If both can be achieved, we can even obtain a full-stack analysis of the behavior of the 3DGS and ADS systems. This is a very interesting and feasible approach, and we are very happy to discuss this route with you.
>
> Weakness 3: The performance improvement appears somewhat incremental. Evaluating the method on additional and more diverse scenarios could better demonstrate its effectiveness and practical significance.
> Question 1: Would an adaptive λ scheduling strategy (balancing Lvisual vs Lperc/Lobj) further improve trade-offs?
>
> Ans: We understand this question is essentially an extension of the "weakness" issue, and we'll address it here as well.
>
> Thank you for raising this point; it prompts us to more clearly articulate the core value of our work.
>
> We believe our main contribution lies not in achieving a massive performance boost, but in revealing and resolving a problem with 3DGS for ADS: the gap between visual quality metrics and the needs of downstream tasks. Existing 3DGS research in autonomous driving almost entirely focuses on improving pixel-level performance such as PSNR/SSIM. Here, we consider the output of the perception task as well. In this study, we focus more on identifying this gap and demonstrating that it can be resolved.
>
> Therefore, to more clearly demonstrate the improvements achievable with a perception-aware 3DGS method, we've abandoned detailed parameter tuning methods and simply pointed out that considering the perception output—that is, applying our two improvements—can improve perception stability, thereby benefiting downstream tasks.
>
> As you pointed out, adaptive lambda can undoubtedly be further improved! More refined loss design and other optimizations also have the potential to improve this metric. We hope that future work can delve deeper into this issue and figure out what optimizations can more effectively improve perception stability.

---

> ### Comment · Reviewer_sxkk · 2025-11-20
> **Reply to the rebuttal**
>
> Thanks for the rebuttal.
>
> Although this paper makes sense, the writing and figures need to be refined. For example, in the conclusion section, some typos still appear in the current version of paper, such as "aiming to ensur e 3 DGS preserve ". I do think these minors can be reviesed but not simply state " indeed found some issues".
>
> Thus I remain the current score.

---

> > ### Author Response · Authors · 2025-11-26
> >
> > Thank you very much for recognizing our work and for carefully pointing out the issues in our writing and figures. We truly appreciate your detailed comments — they are very valuable for improving the overall quality of the paper.
> >
> > We apologize for the remaining typos and expression issues, such as “aiming to ensur e 3 DGS preserve”, and we will carefully refine the writing and revise all such problems in the updated refined version of the PDF. The refined version is currently awaiting the completion of several additional experiments, and we will release it as soon as they finish. Once uploaded, we will promptly reply directly to this comment.
> >
> > Thank you again for your time and constructive feedback.

---

### Official Review · Reviewer_45Bt · 2025-11-01

**Soundness:** 3
**Presentation:** 3
**Contribution:** 3
**Rating:** 6
**Confidence:** 4

**Summary:**

This paper identifies that existing 3D Gaussian Splatting (3DGS) methods for autonomous driving focus on visual quality but neglect the generation performance regarding to downstream perception task.. To address this, the authors propose two simple yet effective losses: (1) Perception-aligned loss, aligning reconstruction with detection results, and (2) Object-zone quality loss, emphasizing reconstruction quality on detected object regions. Experiments on Waymo scenes using S3Gaussian, OmniRe, and EMD show improved mAP, mean IoU, and fewer missed detections, while maintaining similar SSIM/PSNR.

**Strengths:**

Important and Well-Motivated Problem: The paper clearly identifies a fundamental limitation in current 3D reconstruction pipelines for autonomous driving simulation. The proposed concept of perception stability offers a more meaningful and task-relevant objective than traditional visual fidelity metrics.

Simple, Efficient, and Effective Solution: The proposed loss term, $\mathcal{L}_{\text{obj-vis}}$, is intuitive, easy to integrate, and incurs minimal computational overhead, yet consistently yields substantial gains in perception stability and reconstruction quality.

**Weaknesses:**

+ Model-Specific Limitation:
The framework’s dependence on a single “ground-truth” perception model ($\mathcal{P}$) may restrict the generality of the reconstructed scenes, potentially reducing their usefulness for training or evaluating diverse perception architectures.

+ Limited Perception Scope:
The study defines “perception” solely in terms of 2D object detection, overlooking other essential autonomous driving tasks such as 3D detection, semantic segmentation, and multi-object tracking.

**Questions:**

As discussed in the weakness:

+ Generalization:
How does $\mathcal{L}_{\text{obj-vis}}$ perform when trained with one detector (e.g., YOLO) but evaluated using a fundamentally different model family (e.g., DETR-based architectures or 3D detectors)?

+ Task Scope:
Have you explored extending the framework to other perception tasks—for example, leveraging semantic segmentation masks to emphasize safety-critical pixels? Additionally, does the proposed approach improve temporal stability or downstream planning performance in autonomous driving systems?

---

> ### Author Response · Authors · 2025-11-18
>
> Thank you for your affirmation of our work motivation and effectiveness; we are deeply encouraged. At the same time, your two questions regarding generalization and task scope are very important, and we will address them in detail here.
>
> 1.	Generalization: How does perform when trained with one detector (e.g., YOLO) but evaluated using a fundamentally different model family (e.g., DETR-based architectures or 3D detectors)?
>
> Ans: This is an excellent suggestion, and we immediately conducted supplementary experiments. We used the YOLO results to assist the training results and evaluated them using RT-DETR. Taking EMD (S3G) as an example, the results are as follows:
>
> &   mAP   & meanIOU & map   & meanIOU & Miss \\
>
> EMD($S^3G$) & 0.923 & 0.855   & 0.518 & 0.770   & 0    \\
>
> EMD($S^3G$)+ $\mathcal{L}_\text{perc}$       & 0.951 & 0.923   & 0.674 & 0.875   & 0    \\
>
>  EMD($S^3G$)+ $\mathcal{L}_\text{obj-vis}$           & 0.952 & 0.946   & 0.666 & 0.870   & 0
>
> More results will be provided in the revised PDF later. The data shows that the conclusion remains unchanged after using the DETR architecture detector: our method can achieve higher perception stability without reducing pixel-level performance, and this perception stability is not dependent on specific detector.
>
> 2.	Task Scope: Have you explored extending the framework to other perception tasks—for example, leveraging semantic segmentation masks to emphasize safety-critical pixels? Additionally, does the proposed approach improve temporal stability or downstream planning performance in autonomous driving systems?
>
> Ans：Thank you for pointing out this important direction.
>
> Regarding other perceptual tasks (such as semantic segmentation): In fact, our Object-zone Quality Loss is conceptually very close to a foreground-focused semantic segmentation loss designed for the "object" category. Extending the framework to full semantic segmentation is a direct and very promising extension. To achieve this, it might be possible to introduce a segmentation model to generate ground truth segmentation ($S_{gt}$) and design a similar Perception-aligned Loss to align the rendered image with the segmentation results of $S_{gt}$. We believe this would help improve the reconstruction quality of static elements such as roads, sidewalks, and traffic signs.
>
> Regarding temporal stability and planning performance: This is one of the important directions for future work. Perceptual stability is a prerequisite for temporal stability and reliable planning. Current work focuses on solving the perceptual jitter problem at the single-frame level. Naively speaking, a more stable and accurate scene reconstruction, after temporal integration, will help generate smoother target trajectories, which will benefit downstream tracking and planning modules. However, as we found in this study, high similarity does not necessarily lead to high perceptual stability, and we also cannot say that higher reconstruction stability will lead to better temporal stability and planning performance. This requires further investigation. At the very least, the conclusions of this paper can serve as a starting point for future work, and we believe that the methodology presented in this paper can also be transferred to subsequent work.

---

### Official Review · Reviewer_ei8B · 2025-11-01

**Soundness:** 3
**Presentation:** 3
**Contribution:** 3
**Rating:** 6
**Confidence:** 4

**Summary:**

This paper points out that existing 3D Gaussian Splatting methods focus on global visual fidelity, which does not necessarily preserve the perception outputs of autonomous-driving detectors. To address this, the authors introduce two simple training objectives—a perception-aligned loss and an object-zone quality loss—that encourage consistency between reconstructed and original images at the detector level. Experiments on Waymo scenes show improved detection consistency (mAP/IoU/Miss) over several 3DGS baselines without degrading visual quality.

**Strengths:**

1. The insight that “high SSIM ≠ stable perception” is novel and meaningful. Bridging 3DGS reconstruction and detector consistency provides a new angle on how generative scene reconstructions might better serve ADS evaluation.

2.  Despite the conceptual simplicity, the proposed objectives yield measurable mAP/IoU gains across multiple 3DGS baselines and even transfer to unseen detector architectures, indicating that the effect is real and not detector-specific.

3. The paper is concise, well-structured, and easy to follow; figures and tables are clear, and the motivation flows logically from observation → method → experiment.

**Weaknesses:**

1. Limited evidence for “visual-quality ≠ perception.” The claim is mainly supported by SSIM vs mAP differences in one table. Including PSNR, LPIPS, or correlation plots would make the observation statistically more convincing.

2. Running a full YOLO inference at every iteration introduces heavy overhead. While the authors mention this qualitatively, no quantitative efficiency table or training-time comparison is given.

3. The work evaluates only object detection, whereas ADS perception also involves segmentation, tracking, and planning. Without any closed-loop or end-to-end evidence, it remains unclear how perception-stability improvements benefit actual ADS behaviour.

4. The perception-aligned loss is computed after a detector that includes NMS and hard thresholding, which are non-differentiable. The paper does not explain how gradients flow through this step or whether only pre-NMS continuous outputs are used. As written, the formulation is theoretically questionable.

**Questions:**

Please clarify the gradient pathway in the perception-aligned loss: is NMS bypassed, and how are matched detections determined when counts differ?

To substantiate the “visual ≠ perception” claim, report correlations among SSIM, PSNR, LPIPS, and mAP, or plot them across scenes.

Add an efficiency table (e.g., training time per 1k frames or per epoch) comparing baseline 3DGS, + L_perc, and + L_obj.

Consider a lightweight feature-alignment proxy (e.g., backbone-feature L2/KL) to avoid full detector forward passes.

If possible, include a simple downstream experiment  to show whether perception-stable reconstructions yield tangible ADS gains.

---

> ### Author Response · Authors · 2025-11-18
>
> We sincerely thank the reviewer for the detailed and constructive feedback. We are glad that the reviewer finds our insights novel, our motivation clear, and our improvements. Below we address each concern point-by-point and will incorporate corresponding clarifications in the revision.
>
> Question 1:
> 1.	Please clarify the gradient pathway in the perception-aligned loss: is NMS bypassed, and how are matched detections determined when counts differ?
>
> Ans: We thank the reviewer for pointing this out. Our method does not backpropagate through the detector, and therefore does not require gradients to pass through NMS or thresholding. The detector is completely frozen, and is only used to provide a forward mapping from the rendered image to bounding boxes. Consequently, NMS does not lie on the gradient pathway.
> The gradient pathway is: 3DGS parameters → rendered image → IoU-based loss → 3DGS parameters.
> In this way, the detector is fully outside this computational graph, and its outputs are treated as constants t o 3D GS part. This is why no gradients are required to flow through NMS. As long as the rendered image is differentiable with respect to the 3DGS parameters (which it is), the IoU loss remains fully differentiable.
> In subsequent revisions to the article, we will emphasize that the detector is a separate module from the 3DGS process within frozen, to avoid any misunderstanding.
>
> 2.	To substantiate the “visual ≠ perception” claim, report correlations among SSIM, PSNR, LPIPS, and mAP, or plot them across scenes.
> Ans: Thank you for your reminder; this was indeed a missing experimental analysis. Therefore, we added PSNR and LPIPS calculations for all experimental results and performed Pearson Correlation Coefficient analysis. We found that the absolute value of the Pearson correlations between maP and SSIM, PSNR, and LPIPS mostly fell within the range of 0.3-0.6. Taking EMD (S3G) as an example, its Pearson correlations with mAP(yolo) were 0.374, 0.319, and -0.389（LPIPS lower is better）, respectively; Person correlations with mAP(Fasterrcnn) were 0.504, 0.482, and -0.579, respectively.（ p <0.01 in all these stats） In other words, the correlation between pixel-level image quality and object detection stability was only weak-to-meduim in strength. Complete results will be updated in the revised version.
>
> 3.	Add an efficiency table (e.g., training time per 1k frames or per epoch) comparing baseline 3DGS, + L_perc, and + L_obj.
>
> Ans: In section 7.3 of the paper, we did only compare the total runtime. Here, and in subsequent revisions, we show the computational speed of different methods after the Gaussian point split/drop has been frozen.
>
> The data below is the average computation time per 100 epochs (in seconds):
>
> & origin & +$L_{perc}$  & +$L_{obj-vis}$
>
> S3G           & 25.20 & 26.67 & 25.30 \\
>
> OmniRe        & 34.84 & 36.07 & 34.91 \\
>
> EMD(S3G)      & 32.44 & 33.94 & 32.51 \\
>
> EMD(OmniRe)   & 44.94 & 46.45 & 45.00 \\
>
> The data shows that L_{perc} does indeed slightly increase computation time, while L_{obj} adds almost no additional computational overhead. This aligns with the basic conclusions in Fig. 4 of the paper.
>
> 4.	Consider a lightweight feature-alignment proxy (e.g., backbone-feature L2/KL) to avoid full detector forward passes.
>
> Ans: This is really a valuable suggestion. Using early-layer backbone features indeed provides a reasonable lightweight alternative, and we have already started running experiments that align the first five backbone layers using a KL-divergence feature loss. Once the results are completed, we will immediately update here.
>
> 5.	If possible, include a simple downstream experiment to show whether perception-stable reconstructions yield tangible ADS gains.
>
>
> Ans: We totally agree that this is the ultimate goal of this research path. Our core contribution in this work lies in proposing a new and more relevant dimension for reconstruction evaluation—perception stability. Its input (RGB Domain) and output (perception results) are only connected to the perception model itself, thus allowing for direct correlation and optimization. However, considering downstream modules raises the issue of model-to-model interaction. For example, does the downstream model have a tolerance range that allows for some error in the upstream model without causing output errors in the downstream model, or allows the downstream model to amplify the error of the upstream model? This is highly dependent on the capabilities of the downstream model and the overall architecture of the ADS. Considering that even the simplest non-End2End ADS architecture consists of several modules—Perception, Prediction, Planning, and Control—its analysis will be a systematic project. This may be far beyond the scope of this work, but it is the goal we hope to achieve. If there is an opportunity in the future, we hope to achieve this goal by referring to work on module interaction in the ADS field.

---

> > ### Comment · Reviewer_ei8B · 2025-11-26
> >
> > The authors have addressed my main concerns, and the additional analyses strengthen the submission. I am therefore raising my score.

---

> > > ### Author Response · Authors · 2025-11-27
> > >
> > > Thank you very much for recognizing our work and for your comments.
> > >
> > > Your suggestions have played a crucial role in enhancing the completeness of this work, making the discussions on the relationship between pixel-level quality and perception stability, the role of the loss function, the analysis and optimization of computational cost, and future work more comprehensive.
> > >
> > > We will complete the supplementary experiments and prepare a refined version within the next few days.
> > >
> > > Once again, we sincerely appreciate your professional and patient review.

---

### Author Response · Authors · 2025-12-03
**Summary Response from Authors**

We sincerely thank all reviewers for their hard work, which has been extremely helpful in improving the quality of our paper. We are especially grateful to Reviewer ei8B for acknowledging our discussion in the early stage and for the score adjustment, and to Reviewer sxkk for recognizing that the paper content makes sense (even though the score not be updated due to the weakness of typos in origin version PDF). Unfortunately, we did not have the opportunity for further discussion later on; here, we would like to summarize the improvements we have made.

In this refined version, we have mainly made the following modifications(text with blue color in refined version PDF):

1.	Added PSNR and LPIPS as pixel-level metrics to better reflect pixel-level quality (Reviewers ei8B, 45Bt).
2.	Reported correlations analysis results between pixel-level metrics and perception quality, providing stronger evidence for the gap that “high visual similarity ≠ reconstruction stability” (Reviewer ei8B).
3.	Add more time cost analysis (Reviewer ei8B)
4.	Included evaluation using the RT-DETR model to demonstrate that improvements in reconstruction stability are not perception-model-specific (Reviewer 45Bt, 7ST4).
5.	Added experiments and analysis with dual-loss optimization to further validate our approach (Reviewer 7ST4).
6.	Emphasize that the IoU-based loss used in our method does not propagate back to the detector itself(Reviewer ei8B)
7.	Corrected typos (Reviewer sxkk).

Due to page limits, not all discussions with the reviewers could be included in the refined version. We are grateful to Reviewers ei8B, 45Bt, and sxkk for noting that our method can be extended to downstream perception tasks in ADS, and even to other domains, to improve the reliability of 3DGS in autonomous driving scenario generation. This is precisely the direction we plan to pursue in future work, and we hope that the discussions can provide useful insights for further research. We also thank Reviewers sxkk and 7ST4 for suggesting more detailed analysis of the results, particularly regarding trade-offs. As discussed, we believe that in some scenarios, a slight reduction in visual quality is worthwhile to achieve improved perception stability in most scenarios. This aligns with Reviewer sxkk’s point about the choice of $\lambda$: we currently use a simple uniform weighting, and a dynamic weighting strategy could likely further improve the method quantitatively.

In summary, we sincerely thank all reviewers once again; it is thanks to your valuable feedback that this paper has been significantly improved.

---

### Meta-Review · Area_Chair_PdEW · 2026-01-05

**Summary:**

This paper identify and addresses the gap between traditional visual-fidelity metrics and perception stability for 3D Gaussian Splatting in autonomous driving. The proposed perception-aligned and object-zone quality losses improve detection consistency across multiple 3DGS baselines without degrading visual quality. However, some reviewers noted the performance gains as incremental.

The authors clarified the gradient pathway, provided quantitative correlations between pixel-level and perception metrics, added computational cost analysis, and showed generalization to a DETR-based detector. After the rebuttal, several reviewers acknowledged the strengthened submission. The work provides a foundation for future research into task-aware reconstruction. However, the work does not yet validate its core premise with experiments on downstream tasks (e.g., tracking, planning) or closed-loop ADS behavior. The practical benefit of perception-stable reconstructions for the broader autonomous driving pipeline is not yet experimentally substantiated.

In summary, I believe the authors have addressed the main concerns in the rebuttal, and some reviewers are willing to raise their scores. The remaining points could be further polished in the final revision or suitably framed as future work.

**Reviewer Concerns:**

See above.

**Reviewer Scores:**

While some reviewers may raise their scores, others (like the one who gave the lowest rating) may not.

---

### Decision · Program_Chairs · 2026-01-26

Accept (Poster)